# The relationship between sarcopenic obesity and cognitive functionality among inpatients with stable schizophrenia

Yan Guo[1◉], Jianfei Wu[2◉], Xiuping Lei[1], Hongli Zhang[1], Binyou Wang[2], Yu Liu[2], Maoya Xu[2], Yilin Wang◉[2]*, Youguo Tan[1,2]*

**1** Department of Psychiatry, the Zigong Affiliated Hospital, Southwest Medical University, Zigong, Sichuan, China, **2** Zigong Institute of Brain Science, Zigong Mental Health Center, the Zigong Affiliated Hospital, Southwest Medical University, Zigong, Sichuan, China

◉ These authors contributed equally to this work.
* wangyilinzg0321@163.com (YLW); tanyoug1964@sina.com (YGT)

**Data availability statement:** All relevant data are within the manuscript and its Supporting Information files.

**Funding:** This study was supported by the Zigong Key Science and Technology Plan (Collaborative Innovation Project of Zigong

## Abstract

### Background and objectives

Patients with schizophrenia face an elevated risk of sarcopenic obesity (SO) due to antipsychotic-induced metabolic dysfunction, physical inactivity, and nutritional deficiencies. Although recent studies suggest an association between SO and cognitive decline, its specific impact on cognitive function in schizophrenia remains to be fully elucidated. This study aimed to evaluate the diagnostic concordance between the European Society for Clinical Nutrition and Metabolism/European Association for the Study of Obesity ($SO_{ESPEN}$) criteria and its modified version ($SO_{ESPEN-M}$), and to examine their respective associations with cognitive function in inpatients with stable schizophrenia.

### Methods

In this cross-sectional analysis, 228 adults with stable schizophrenia were recruited. SO was diagnosed using two definitions: $SO_{ESPEN}$ (excess adiposity, low muscle mass-to-weight ratio, and reduced handgrip strength) and $SO_{ESPEN-M}$ (BMI-adjusted muscle mass threshold). Cognitive function was assessed using the Montreal Cognitive Assessment-Chinese version (MoCA-C). Multivariate linear regression models were employed to evaluate associations between SO and MoCA-C scores, adjusting for relevant demographic, clinical, and comorbidity-related variables.

### Results

SO prevalence was 17.1% under both diagnostic criteria, with moderate inter-criteria agreement ($\kappa = 0.660$). Sex-stratified analyses revealed divergent diagnostic trends: in males, SO prevalence increased from 15.9% ($SO_{ESPEN}$) to 22.5% ($SO_{ESPEN-M}$;

Institute of Brain Sciences) (Grant Numbers: 2023-NKY-02-04, 2023-NKY-02-07, 2023-NKY-03-03), and the Collaborative Innovation Project of Zigong Medical Big Data and Artificial Intelligence Research Institute (Grant Number: 2024-YGY-02-04). The funder had no role in the design, data collection, data analysis, and reporting of this study.

**Competing interests:** The authors have declared that no competing interests exist.

$\kappa = 0.698$); in females, prevalence decreased from 18.9% to 8.9% ($\kappa = 0.590$). Across both criteria, SO groups demonstrated significantly lower MoCA-C scores (males: 16 vs 20, $p = 0.045$ for $SO_{ESPEN}$; 13 vs 21, $p < 0.001$ for $SO_{ESPEN-M}$; females: 11 vs 17, $p = 0.009$ for $SO_{ESPEN}$; 10.5 vs 17, $p = 0.036$ for $SO_{ESPEN-M}$). Multivariate analysis confirmed that $SO_{ESPEN-M}$-defined SO was independently associated with lower MoCA-C scores in males ($\beta = -2.71$, 95% CI: $-5.08$ to $-0.33$, $p = 0.027$).

## Conclusion

Our results demonstrate that SO defined by $SO_{ESPEN-M}$ criteria is significantly associated with cognitive impairment in male inpatients with stable schizophrenia.

## Introduction

Schizophrenia is a severe neuropsychiatric disorder characterized by a combination of positive symptoms (such as hallucinations or delusions) and negative symptoms (such as affective blunting or social withdrawal). It affects an estimated 23.6 million people worldwide, with a median onset age of 25 years and peak incidence at approximately 20.5 years [1]. Beyond its hallmark psychopathology, schizophrenia is often accompanied by progressively declining cognitive function [2], impaired synaptic function [3], and an elevated risk of metabolic comorbidities [4], all contributing to significant occupational impairment and a high degree of socioeconomic burden [5].

Long-term management of schizophrenia primarily involves pharmacotherapy, especially second-generation antipsychotics (SGAs), as well as psychological interventions, such as cognitive remediation therapy (CRT) and cognitive behavioral therapy (CBT), and neuromodulation techniques like repetitive transcranial magnetic stimulation (rTMS) and modified electroconvulsive therapy (mECT) [6,7]. Although SGAs are essential for symptom control, prolonged use is associated with adverse metabolic effects, including obesity and insulin resistance. The prevalence of metabolic syndrome in individuals with schizophrenia is estimated at 32.6%, two to three times higher than in the general population [8]. These metabolic disturbances, compounded by sedentary lifestyles and poor dietary habits, contribute to the development of sarcopenic obesity (SO), a condition characterized by the simultaneous presence of excessive adiposity and reduced skeletal muscle mass (SMM) with functional impairment [9].

SO creates a neuroendocrine environment characterized by chronic inflammation and insulin resistance, both of which have been implicated in cognitive decline through mechanisms such as blood-brain barrier dysfunction and impaired hippocampal neurogenesis [3,10,11]. Moreover, SO is associated with multiple comorbidities, including cancer [12], cardiovascular disease [13], liver disease [14], chronic obstructive pulmonary disease [15], and various orthopedic disorders [16,17]. Early identification of SO phenotypes via body composition analysis and metabolic profiling is therefore crucial for developing targeted therapeutic strategies. However, discrepancies in SO diagnostic criteria, particularly between the consensus definition

by the European Society for Clinical Nutrition and Metabolism/European Association for the Study of Obesity (SO$_{ESPEN}$) and its modified version (SO$_{ESPEN-M}$), impede clinical standardization and risk stratification [18,19]. While SO$_{ESPEN}$ has been validated in predicting fall risk [19], its use in assessing cognitive function remains underexplored. The modified SO$_{ESPEN}$ criteria, which normalize muscle mass by body mass index (BMI) rather than total body weight, may be especially relevant in schizophrenia populations, where antipsychotic-induced weight fluctuations often obscure traditional muscle-to-weight metrics [20,21]. By adjusting for BMI, SO$_{ESPEN-M}$ distinguishes sarcopenia from adiposity-driven weight changes, providing a more accurate measure of muscle insufficiency. This distinction is particularly important as low muscle mass, independent of BMI, has been mechanistically associated with neuroinflammation and insulin resistance in schizophrenia patients.

Based on these considerations, this study aimed to address two key knowledge gaps: (1) evaluating the diagnostic concordance between SO$_{ESPEN}$ and SO$_{ESPEN-M}$ criteria in hospitalized patients with schizophrenia, and (2) investigating the differential associations of SO phenotypes with cognitive function. Current research studies highlight significant sex differences in body composition traits that are important for SO diagnosis, including sexually dimorphic patterns in muscle mass distribution [22], fat deposition [23], and metabolic adaptation [24]. These sex-specific variations may significantly affect SO phenotypes and their cognitive implications. Therefore, sex-stratified analyses were conducted to examine potential effect modification by sex, intending to generate biologically plausible and clinically actionable insights. By integrating body composition assessments, handgrip strength (HGS) measurement, and cognitive function testing, this study intends to improve the diagnostic precision of SO criteria and identify high-risk schizophrenia subgroups, therefore facilitating the development of targeted interventions aimed at mitigating cognitive decline in this vulnerable population.

## Materials and methods

### Informed consent and ethical approval process

This secondary analysis utilized de-identified data from a 2023 cohort study approved by the Zigong Mental Health Center IRB (2023024), which complied with the Declaration of Helsinki. All original participants provided written informed consent after undergoing a standardized capacity assessment: two psychiatrists verified their comprehension of study objectives, risks, and benefits using ICD-10-based criteria for stable schizophrenia (≥1-month clinical stability). Only adults aged 18 years or older were eligible for inclusion. The IRB waived ethical review for this re-analysis because (1) original consent included authorization for future research use of anonymized data, (2) no further interventions were conducted, and (3) all identifying information was removed before analysis.

### Patients' characteristics

From an initial cohort of 325 individuals, patients with schizophrenia confirmed via ICD-10 diagnostic criteria and dual psychiatric evaluation were consecutively enrolled in August 2023. To ensure clinical stability, participants were required to (a) maintain an unchanged clinical status for at least one month and (b) receive stable pharmacotherapy for a minimum of two months before enrollment [25]. Inclusion criteria were: (1) age ≥ 18 years; (2) a confirmed diagnosis of stable schizophrenia by two psychiatrists; (3) ability to provide informed consent; and (4) completion of all required assessments. Exclusion criteria were: (1) hepatic or renal impairment, (2) autoimmune disease, (3) active oncologic treatment, (4) diagnostic ambiguity regarding schizophrenia stability, (5) incomplete biometric or HGS data, and (6) withdrawal of consent.

### SO characterization

SO was diagnosed using modified SO$_{ESPEN}$ criteria tailored for Asian populations, employing two distinct operational definitions [26]. The SO$_{ESPEN}$ criteria required three elements: (1) excessive adiposity (fat mass >20.21% in males or >31.7% in females), (2) low SMM/body weight <38.2% in males or <32.2% in females), and (3) compromised muscle function (HGS < 28 kg in males or <18 kg in females). The SO$_{ESPEN-M}$ criteria incorporated BMI-adjusted muscle mass thresholds

(SMM/BMI < 1.017 in males/ < 0.677 in females) while retaining the same adiposity and HGS thresholds. Body composition was assessed using the InBody 770 bioelectrical impedance analysis (BIA) device (Biospace, Korea), following manufacturer-recommended standardized protocols.

## Cognitive evaluation

The Montreal Cognitive Assessment (MoCA) scale is a widely validated tool for evaluating cognitive function in individuals with cognitive impairments [27], including those diagnosed with schizophrenia [28]. In this study, trained psychiatrists administered the Chinese version of the MoCA (MoCA-C) version following standardized assessment procedures [29]. The MoCA-C is a 30-point scale evaluating multiple cognitive domains and requires approximately 15 minutes to complete. Lower scores reflect greater cognitive impairment.

## Covariates

A comprehensive set of covariates was systematically collected from multiple data sources. Demographic information, including age, sex, education level (illiterate/high school and below/university and above), marital status (married/unmarried/divorced/widowed), number of siblings, number of children, and smoking and drinking history, was obtained from clinical records. Other clinical variables included vision and hearing impairments, disease duration, hospitalized time, family psychiatric history, first-episode status, fall history, COVID-19 history, number of chronic comorbidities, current antipsychotic regimen, chlorpromazine equivalent dose, and relevant metabolic and insulin resistance markers (fasting blood glucose; alanine aminotransferase [ALT]/aspartate aminotransferase [AST] ratio; uric acid; triglyceride[TG]/high-density lipoprotein cholesterol [HDL-C] ratio). Psychological variables included depression symptoms assessed using the Patient Health Questionnaire-9 (PHQ-9) scale [30], anxiety symptoms using the Generalized Anxiety Disorder-7 scale (GAD-7) [31], and psychotic symptoms using the Positive and Negative Syndrome Scale (PANSS) [32]. For female participants, further covariates included age at menarche, menopause status, and age at menopause. Anthropometric measurements, including height, weight, and HGS, were obtained using calibrated equipment under standardized conditions. The HGS assessment protocol has been described previously [33]. Data collection followed a structured sequence: variables from electronic medical records were first extracted, followed by anthropometric measurements, and then psychometric evaluations. Quality control protocols were rigorously implemented by trained research staff, achieving over 95% completeness rate in the final dataset and eliminating the need for missing data imputation.

## Statistical analyses

All statistical analyses were conducted using SPSS 25.0 (IBM Corp) following a dual analytical strategy. Descriptive analyses were used to summarize nominal variables as counts and percentages, while continuous variables were reported as medians (P25, P75) due to their non-normal distribution. Comparative analyses were performed using non-parametric tests, including the Mann-Whitney U test and the Kruskal-Wallis H tests, to evaluate differences in cognitive scores across SO phenotypes.

To assess the concordance between SO classification criteria (SO$_{ESPEN}$ vs. SO$_{ESPEN-M}$), Cohen's kappa ($\kappa$) coefficient was calculated, with interpretation based on the classification framework proposed by McHugh (2012) [34]. Associations between SO classification and cognitive function (MoCA-C scores) were analyzed using multivariate linear regression models adjusted for covariates. Covariates were selected in two steps: first, univariate analyses identified variables significantly associated with MoCA-C scores ($p < 0.05$); second, these variables were incorporated into the fully adjusted model.

For males, the final model included age, disease duration, education level, hearing impairment, drinking history, PANSS-negative symptom score, PANSS-general psychopathology score, chlorpromazine equivalent dose, and ALT/AST ratio. For females, the adjusted model included age, number of siblings and children, education level, history of falls, PANSS-negative symptom score, PANSS-general psychopathology score, and menopause status.

Data visualization was performed using matplotlib (version 3.10.0) in Python and an online website (https://www.med-sta.cn/software) for generating box plots and multi-factor linear regression forest plots, respectively.

## Results

### Participant characteristics

Fig 1 illustrates the participant selection flowchart. From an initial pool of 325 hospitalized individuals with schizophrenia, 228 patients met the inclusion criteria following the application of exclusion criteria. Participants were excluded for the following reasons: age < 18 years (n = 12), presence of severe hepatic or renal dysfunction or ongoing oncologic treatment (n = 20), incomplete HGS data (n = 20), clinical instability (n = 8), refusal to participate (n = 30), and missing biometric data (n = 7).

Table 1 presents the demographic and clinical characteristics of the final cohort, stratified by sex (male: n = 138; female: n = 90). Significant sex-based differences were observed. Compared to males, females demonstrated higher rates of marriage (32.2% vs. 10.9%), family history of psychiatric disorders (30.0% vs. 18.1%), and multimorbidity defined as ≥2 chronic conditions (23.3% vs. 9.4%). Similarly, males showed significantly higher rates of smoking (64.5% vs. 7.8%) and alcohol use (40.6% vs. 3.3%). Certain parameters were comparable across sexes, including educational level (≥high school: 94.2% in males vs. 90.0% in females), predominant use of atypical or combined antipsychotic regimens (97.1% vs. 98.9%), and median disease duration (~20 years). Analysis of continuous variables revealed metabolic differences; females had higher BMI (25.4 vs. 24.1 kg/m²), whereas males showed higher ALT/AST ratios (0.95 vs. 0.90). Psycho-pathological indices also varied: females scored higher on PANSS-negative symptoms (19 vs. 17) and PANSS total scores (64.5 vs. 59), while males had higher cognitive scores (MoCA-C: 20 vs. 16).

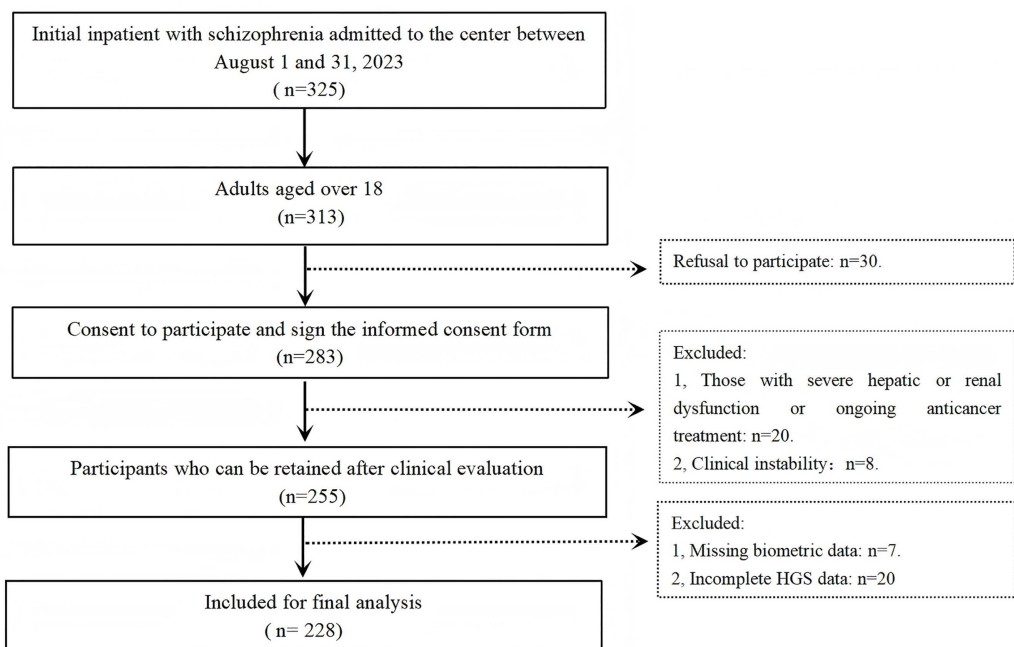

**Fig1. Flow-chart of the participants selection process.** The number of participants excluded or maintained at each step is reported with the related reasons. HGS, handgrip strength.

**Table 1. General information of enrolled inpatients with stable schizophrenia.**

| Variable | Male, n=138 | Female, n=90 |
|---|---|---|
| **Age(years), median(P25,P75)** | 50.5(41,55) | 51(40,58) |
| **BMI(kg/m², median(P25,P75)** | 24.1(21.8,27.1) | 25.4(23.2,28.8) |
| **Marital status, n(%)** | | |
| Married | 15(10.9) | 29(32.2) |
| Unmarried/divorced/ widowed | 123(89.1) | 61(67.8) |
| **Education, n(%)** | | |
| Illiterate | 7(5.1) | 9(10.0) |
| High school and below | 119(86.2) | 70(77.8) |
| University and above | 11(8.0) | 11(12.2) |
| **Disease duration (years), median(P25,P75)** | 20(12,28) | 18.5(9,29.3) |
| **Hospitalized time(months), median(P25,P75)** | 29(9.8,54.3) | 20(6,44) |
| **Number of siblings, median(P25,P75)** | 3(2,4) | 4(2,5) |
| **Number of children, median(P25,P75)** | 0(0,1) | 1(0,2) |
| **Family history of mental disorder, n(%)** | | |
| No | 112(81.2) | 63(70) |
| Yes | 25(18.1) | 27(30) |
| **First episode, n(%)** | | |
| No | 132(95.7) | 89(98.9) |
| Yes | 6(4.3) | 1(1.1) |
| **Vision problems, n(%)** | | |
| No | 125(90.6) | 76(84.4) |
| Yes | 13(9.4) | 14(15.6) |
| **Hearing problems, n(%)** | | |
| No | 128(92.8) | 83(92.2) |
| Yes | 10(7.2) | 7(7.8) |
| **Smoking history, n(%)** | | |
| No | 49(35.5) | 83(92.2) |
| Yes | 89(64.5) | 7(7.8) |
| **Drinking history, n(%)** | | |
| No | 82(59.4) | 87(96.7) |
| Yes | 56(40.6) | 3(3.3) |
| **Falls history, n(%)** | | |
| No | 133(96.4) | 82(91.1) |
| Yes | 5(3.6) | 8(8.9) |
| **COVID-19 history, n(%)** | | |
| No | 83(60.1) | 53(58.9) |
| Yes | 53(38.4) | 37(41.1) |
| **Number of chronic diseases, n(%)** | | |
| 0 | 97(70.3) | 58(64.4) |
| 1 | 28(20.3) | 11(12.2) |
| ≥2 | 13(9.4) | 21(23.3) |
| **Antipsychotics, n(%)** | | |
| Typical | 4(2.9) | 1(1.1) |
| Atypical | 128(92.8) | 80(88.9) |
| Combined | 6(4.3) | 9(10) |

*(Continued)*

**Table 1.** (Continued)

| Variable | Male, n=138 | Female, n=90 |
|---|---|---|
| **Chlorpromazine equivalent dose (mg/day), median(P25,P75)** | 300(200,525) | 375(200,600) |
| **ALT/AST,median(P25,P75)** | 0.95(0.70,1.20) | 0.90(0.68,1.13) |
| **TG/HDL-C, median(P25,P75)** | 1.52(0.79,2.13) | 1.21(0.72,1.92) |
| **Uric acid(umol/L), median(P25,P75)** | 366.1(311.6,408.85) | 299.5(257.6,386.7) |
| **Fasting blood glucose (mmol/L), median(P25,P75)** | 4.97(4.59,5.41) | 5.30(4.83,5.88) |
| **PANSS profile-total score, median(P25,P75)** | 59(50,69) | 64.5(55.8,75.3) |
| **PANSS profile-positive symptoms, median(P25,P75)** | 13(9,16) | 13.0(8.8,19.0) |
| **PANSS profile-negative symptoms, median(P25,P75)** | 17(13,21) | 19(16,24) |
| **PANSS profile-general psychopathology, median(P25,P75)** | 29(24,33) | 30(27,36) |
| **MoCA-C scores, median(P25,P75)** | 20(14,24) | 16(9,21) |
| **PHQ-9 Scores, median(P25,P75)** | 2.5(1.0,6.0) | 3.0(1.0,4.3) |
| **GAD-7 Scores, median(P25,P75)** | 0(0,3) | 1(0,3) |
| **Age at menarche(years), median(P25,P75)** | – | 14(14,15) |
| **Menopause, n(%)** | | |
| No | – | 37(41.1) |
| Yes | – | 52(57.8) |
| **Age at menopause(years), median(P25,P75)** | – | 50(47,50) |

Note: ALT, Alanine Aminotransferase; AST, Aspartate Aminotransferase; TG, Triglyceride; HDL-C, High-Density Lipoprotein Cholesterol; PANSS, Positive and Negative Syndrome Scale; MoCA-C, Montreal Cognitive Assessment-Chinese version; PHQ-9, Patient Health Questionnaire-9; GAD-7, Generalized Anxiety Disorder-7.

## SO classification discrepancies

Table 2 delineates diagnostic concordance between SO$_{ESPEN}$ criteria and SO$_{ESPEN-M}$ criteria for SO in 228 schizophrenia inpatients. Overall, both criteria yielded identical overall SO prevalence (17.1%, n=39/228) with 28 concordant SO cases (12.3%), yielding moderate inter-criteria agreement ($\kappa$=0.660). Sex- specific analyses revealed significantly divergent reclassification patterns. Among males (n=138), SO prevalence increased from 22 (15.9%) under SO$_{ESPEN}$ to 31 (22.5%) under SO$_{ESPEN-M}$, with a $\kappa$ value of 0.698. Among females (n=90), the prevalence decreased from 17 (18.9%) under SO$_{ESPEN}$ to 8 (8.9%) under SO$_{ESPEN-M}$; all 9 reclassified cases shifted status as non-SO, resulting in zero overlapping SO cases under the two criteria ($\kappa$=0.590).

Fig 2 presents MoCA-C scores stratified by SO status and diagnostic criteria, with significant differences observed across all strata (p<0.05). Both the SO$_{ESPEN}$ and SO$_{ESPEN-M}$ criteria consistently revealed significantly lower MoCA-C scores in individuals diagnosed with SO. Under the SO$_{ESPEN}$ framework, SO males had MoCA-C scores 20% lower than non-SO males (16 vs. 20; p=0.045), while SO females showed a 35% reduction (11 vs. 17; p=0.009). The SO$_{ESPEN-M}$ criteria revealed even higher deficits, with SO males and females scoring 13 vs. 21 (p<0.001) and 10.5 vs. 17 (p=0.036), respectively. These findings indicate more pronounced cognitive impairment under the SO$_{ESPEN-M}$ definition, with performance gaps exceeding 35% in both sexes.

## Relationships between SO and cognitive function

Potential covariates for sex-specific model adjustment were identified through univariate regression analyses (Table 3). In males, MoCA-C scores were significantly associated with age, disease duration, education level, hearing impairment, alcohol use history, PANSS negative symptoms, PANSS general psychopathology scores, chlorpromazine equivalent dose, and the ALT/AST ratio (all

**Table 2. Differences among stable schizophrenia inpatients with sarcopenic obesity identified using the ESPEN/EASO criteria and modified versions thereof.**

| Variable | | SO_ESPEN criteria | | | | | | | | |
|---|---|---|---|---|---|---|---|---|---|---|
| | | Total(n = 228) | | | Male(n = 138) | | | Female(n = 90) | | |
| | | Non-SO | SO | Kappa-value | Non-SO | SO | Kappa-value | Non-SO | SO | Kappa-value |
| SO_ESPEN-M criteria | Non-SO | 178 | 11 | 0.660 | 105 | 2 | 0.698 | 73 | 9 | 0.590 |
| | SO | 11 | 28 | | 11 | 20 | | 0 | 8 | |

Note: SO, sarcopenic obesity; SO_ESPEN, ESPEN/EASO criteria; SO_ESPEN-M, ESPEN/EASO criteria modifications.

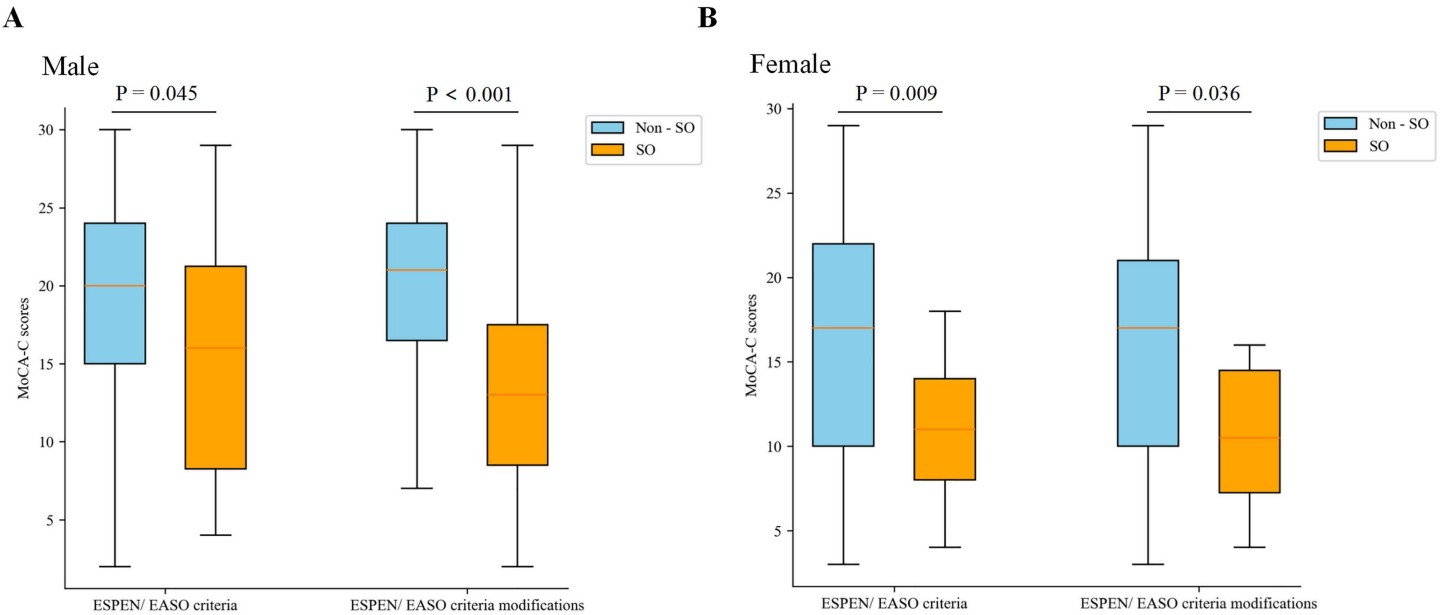

**Fig 2. Comparison of MoCA-C scores between SO and non-SO groups defined by different criteria in males and females.** SO, sarcopenic obesity; MoCA-C, Montreal Cognitive Assessment-Chinese version.

p<0.05). For females, significant associations included age, number of siblings, number of children, education level, fall history, PANSS negative symptoms, PANSS general psychopathology, and menopause status. These factors were retained as candidate covariates in the subsequent sex-stratified multivariate models examining cognitive outcomes in patients with stable schizophrenia.

Fig 3 summarizes the results of the multivariate linear regression analyses. After adjusting for relevant covariates, SO defined by the SO_ESPEN-M criteria was independently associated with significantly lower MoCA-C scores in males ($\beta$ = −2.71, 95% CI: −5.08 to −0.33, p = 0.027). Other variables significantly associated with lower cognitive scores in males included PANSS negative symptom severity ($\beta$ = −0.34, 95% CI: −0.52 to −0.16, p < 0.001), high school and below ($\beta$ = 8.76, 95% CI: 4.36 to 13.17, p < 0.001), university and above ($\beta$ = 11.18, 95% CI: 5.72 to 16.63, p < 0.001), and a history of alcohol consumption ($\beta$ = −2.51, 95% CI: −4.42 to −0.60, p = 0.011).

Among females, SO status defined by either the SO_ESPEN or SO_ESPEN-M criteria was not significantly associated with MoCA-C scores (p > 0.05). Instead, cognitive performance was independently influenced by PANSS negative symptoms, post-menopausal status, and education level (all p < 0.05). No other covariates retained significance in the final adjusted models for females (Fig 3).

**Table 3. Univariate analysis of general information and MoCA-C scores in patients with stable schizophrenia.**

| Variable | Male, n = 138 | | Female, n = 90 | |
|---|---|---|---|---|
| | β(95%CI) | P-value | β(95%CI) | P-value |
| Age | −0.249(−0.355,-0.143) | **<0.001** | −0.175(−0.293,-0.058) | **0.004** |
| BMI | −0.052(−0.364,0.260) | 0.743 | −0.254(−0.603,0.094) | 0.151 |
| Disease duration | −0.137(−0.249,-0.025) | **0.017** | −0.113(−0.233,0.008) | 0.066 |
| Hospitalized time | 0.025(−0.005,0.054) | 0.1 | 0.013(−0.017,0.042) | 0.392 |
| Number of siblings | −0.207(−0.841,0.426) | 0.518 | −0.973(−1.724,-0.222) | **0.012** |
| Number of children | 0.297(−1.879,1.285) | 0.711 | −2.372(−3.843,-0.901) | **0.002** |
| Family history of mental disorder | −1.839(−4.851,1.173) | 0.229 | 2.735(−0.487,5.958) | 0.095 |
| First episode | 1.583(−4.106,7.273) | 0.583 | −3.494(−17.789,10.800) | 0.628 |
| Marital status | 1.367(−2.357,5.092) | 0.469 | 2.555(−0.610,5.719) | 0.112 |
| Education | 6.542(3.502,9.581) | **<0.001** | 7.471(4.706,10.236) | **<0.001** |
| Vision problems | 1.757(−2.208,5.722) | 0.382 | 3.013(−1.077,7.103) | 0.147 |
| Hearing problems | −5.698(−10.073,-1.342) | **0.011** | −3.747(−9.293,1.799) | 0.183 |
| Smoking history | −0.418(−2.844,2.008) | 0.734 | 2.294(−3.287,7.875) | 0.416 |
| Drinking history | −2.440(−4.769,-0.111) | **0.04** | −1.851(−10.200,6.499) | 0.661 |
| Falls history | −4.239(−10.413,1.935) | 0.177 | −7.233(−12.268,-2.177) | **0.006** |
| COVID-19 history | −2.188(−4.574,0.198) | 0.072 | −2.334(−5.343,0.675) | 0.127 |
| PHQ-9 Scores | −0.177(−0.496,0.142) | 0.274 | 0.273(−0.251,0.797) | 0.303 |
| GAD-7 Scores | −0.145(−0.558,0.268) | 0.488 | 0.381(−0.256,1.019) | 0.238 |
| PANSS profile-positive symptoms | −0.012(−0.224,0.200) | 0.914 | −0.024(−0.268,0.219) | 0.842 |
| PANSS profile-negative symptoms | −0.453(−0.608,-0.299) | **<0.001** | −0.632(−0.833,-0.431) | **<0.001** |
| PANSS profile-general psychopathology | −0.282(−0.457,-0.107) | **0.002** | −0.396(−0.605,-0.187) | **<0.001** |
| Number of chronic diseases | −0.989(−2.759,0.782) | 0.271 | −1.523(−3.276,0.23) | 0.088 |
| Antipsychotics | −2.906(−7.198,1.387) | 0.183 | 3.376(−1.240,7.991) | 0.15 |
| Chlorpromazine equivalent dose | 0.005(0.001,0.009) | **0.021** | 0.006(0.000,0.011) | 0.054 |
| Fasting blood glucose | 0.166(−0.902,1.234) | 0.759 | 0.256(−1.000,1.512) | 0.686 |
| ALT/AST | 6.042(2.785,9.299) | **<0.001** | 1.982(−2.463,6.427) | 0.378 |
| TG/HDL-C | −0.048(−0.792,0.696) | 0.899 | 0.088(−1.480,1.656) | 0.911 |
| Uric acid | 0.002(−0.013,0.017) | 0.770 | −0.001(−0.019,0.017) | 0.951 |
| Age at menarche | – | – | −0.05(−1.363,1.262) | 0.939 |
| Menopause status | – | – | −5.966(−8.711,-3.220) | **<0.001** |
| Age at menopause | – | – | 0.262(−0.224,0.749) | 0.284 |

Note: ALT, Alanine Aminotransferase; AST, Aspartate Aminotransferase; TG, Triglyceride; HDL-C, High-Density Lipoprotein Cholesterol; PANSS, Positive and Negative Syndrome Scale; MoCA-C, Montreal Cognitive Assessment-Chinese version; PHQ-9, Patient Health Questionnaire-9; GAD-7, Generalized Anxiety Disorder-7.

## Discussion

This study demonstrates that SO, when defined by the BMI-adjusted SO_ESPEN-M criteria, is independently associated with cognitive impairment in male inpatients with stable schizophrenia, while neither SO_ESPEN nor SO_ESPEN-M classifications showed significant associations in females after adjustment. SO_ESPEN-M identified 41% more males with SO than SO_ESPEN, and this reclassified subgroup exhibited significant cognitive impairment (−38% MoCA-C). The better performance of SO_ESPEN-M in males likely reflects its ability to distinguish sarcopenia from adiposity-related weight fluctuations, a key confounder in antipsychotic-treated populations. These findings suggest that BMI-adjusted

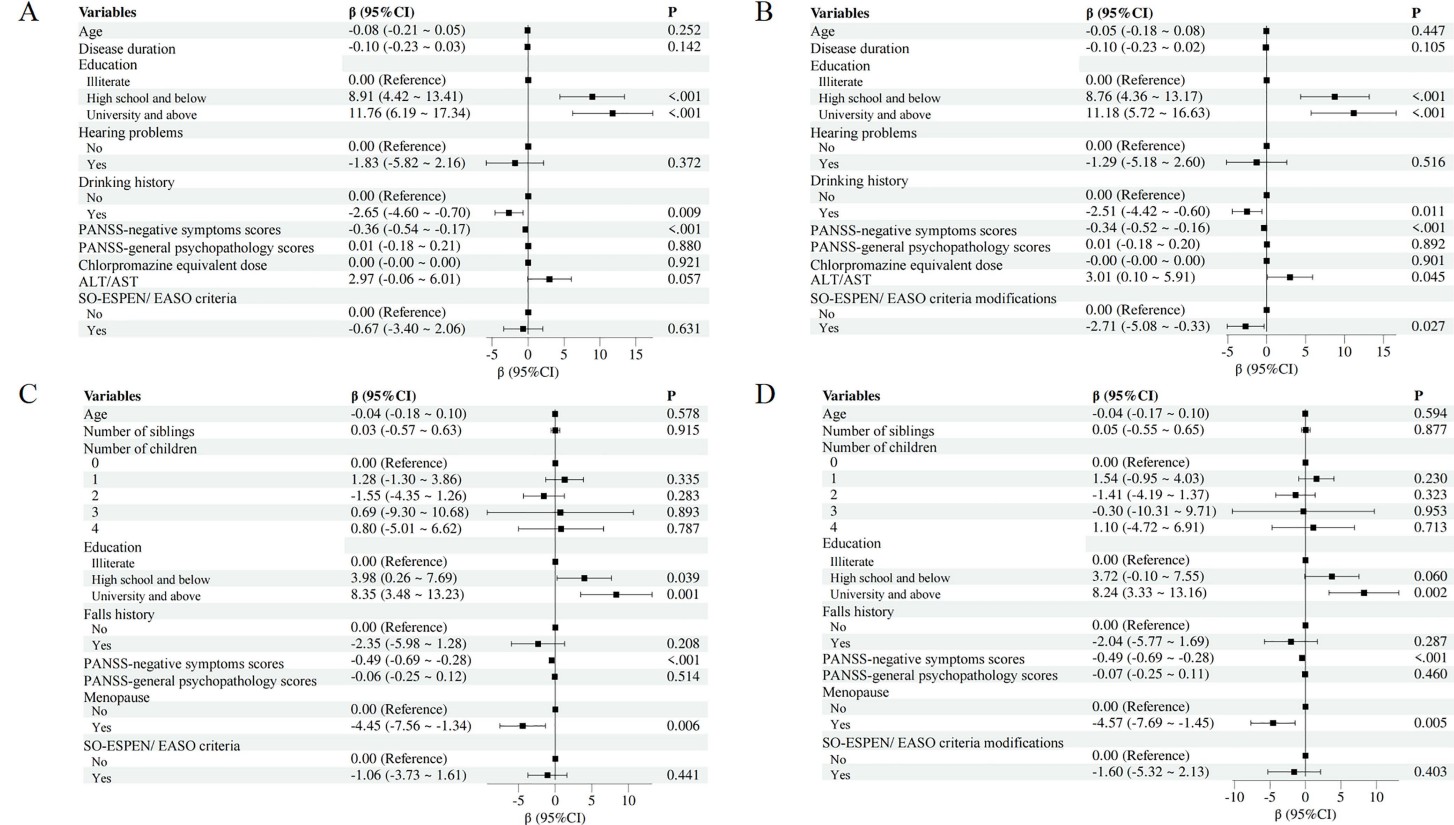

**Fig 3. Forest plot of multivariate linear regression analysis for factors associated with MoCA-C score in schizophrenia inpatients in males (A and B) and females (C and D).** PANSS, Positive and Negative Syndrome Scale; ALT, Alanine Aminotransferase; AST, Aspartate Aminotransferase; SO, sarcopenic obesity.

diagnostics may capture metabolically mediated cognitive vulnerability more accurately in male schizophrenia patients.

As a chronic pathological condition, SO is characterized by a combination of both sarcopenia and pathological adiposity and is mainly diagnosed using the SO$_{ESPEN}$ criteria [35], which have been validated across diverse international populations. For instance, the Italian MoMa study evaluating metabolic syndrome in the context of primary care reported an SO prevalence of 9% [36], while a Brazilian cross-sectional analysis detected a prevalence of 6.5% among older women based on the SO$_{ESPEN}$ criteria [37]. Similarly, a geriatric cohort study conducted in Western China reported prevalence rates of 13.2% (SO$_{ESPEN}$) [26]. This analysis revealed a higher rate of SO prevalence identified using the SO$_{ESPEN}$ criteria than in previous epidemiological reports: 17.1% in total schizophrenia participants, with 15.9% in males, and 18.9% in females. This elevated risk aligns with the clinical characteristics of schizophrenia, which include chronic antipsychotic use, sedentary behavior, social withdrawal, and poor dietary habits, factors that collectively contribute to metabolic vulnerability [38,39]. Moreover, our findings also support the biological plausibility of this association, as the pathogenesis of SO is multifactorial, driven by age-related muscle decline, hormonal disruption, chronic metabolic dysregulation, lifestyle factors, and iatrogenic effects of long-term pharmacotherapy. These factors are also reportedly involved in the pathophysiology of schizophrenia and may contribute to disease progression [18,38,40]. The observed SO prevalence using SO$_{ESPEN}$ (17.1%) was lower than the 23.13% reported in a previous comparative study on hospitalized schizophrenia patients [41]. This discrepancy is likely due to age-related differences, as the previous study included only individuals aged ≥50 years,

whereas our cohort had a median age of 49 years (IQR: 39–54), with 90.8% under 55. Accordingly, our sample likely had shorter disease durations, lower cumulative antipsychotic exposure, and reduced lifestyle-related metabolic burden [38].

The choice between SO diagnostic criteria, normalizing muscle mass by BMI ($SO_{ESPEN-M}$) or total body weight ($SO_{ESPEN}$), significantly alters prevalence estimates and risk stratification, producing significant sex-dimorphic effects. In males, SO prevalence increased by 41% (22.5% vs. 15.9%), while in females it decreased by 53% (8.9% vs. 18.9%). These divergent reclassification patterns likely reflect distinct biological responses to antipsychotic-induced metabolic stress. Cognitive impairment in reclassified $SO_{ESPEN-M}$ males (−38% MoCA-C) affirms the clinical relevance of BMI-adjusted definitions in capturing metabolically driven pathology.

Mechanistically, the stronger association between $SO_{ESPEN-M}$-defined SO and cognitive dysfunction likely reflects its ability to identify a metabolically adverse phenotype characterized by neuroendocrine abnormalities, such as blood-brain barrier disruption mediated by chronic inflammation [42,43], and antipsychotic-exacerbated imbalances between fat and muscle mass [42]. The male-specific association between SO and cognitive impairment may represent a pathophysiological nexus in which adiposity-driven inflammation and myokine dysregulation collectively accelerate neurodegeneration. This process is likely amplified by sex-specific differences in body composition, including divergent muscle distribution and fat deposition patterns [44]. In females, estrogen-mediated myoprotection and menopause-related metabolic shifts [45,46] may modify adiposity-muscle interactions in ways that reduce the sensitivity of BMI-adjusted criteria. This likely accounts for the near-zero diagnostic concordance between $SO_{ESPEN}$ and $SO_{ESPEN-M}$ ($\kappa = 0.590$) and the lack of significant associations with cognitive outcomes in females. Instead, menopause status and PANSS negative symptom scores emerged as dominant contributors to cognitive variability, indicating that SO's effects may be masked by stronger neuroendocrine and psychopathological factors in hospitalized female patients. Accordingly, applying $SO_{ESPEN-M}$ in clinical settings may yield actionable advantages by identifying male patients at elevated cognitive risk, thus facilitating targeted interventions such as resistance training to alleviate sarcopenia. At the same time, these findings emphasize the need for better-suited female-specific diagnostic refinements.

This study has several limitations that warrant consideration. First, its cross-sectional design, although based on validated methods (e.g., MoCA-C, $SO_{ESPEN}$ criteria, BIA), precludes causal inference between SO and cognitive impairment. Longitudinal studies are necessary to determine temporal relationships and evaluate whether SO precedes or results from cognitive decline. Second, the single-center design and limited sample size imposed constraints. Specifically: 1) The sample size, compounded by the scarcity of research on this specific issue in schizophrenia patients, precluded further stratified analyses by specific age groups. Although age and menopausal status were included as covariates, statistical power was insufficient to detect potentially nuanced age-related effects or interactions within narrower age strata. 2) Substantial heterogeneity in antipsychotic medication regimens among participants – with over ten distinct monotherapies or bivalent combinations represented – prevented robust evaluation of potential differential effects of specific antipsychotic types on SO or cognitive outcomes. Third, our exclusive focus on inpatients limits the generalizability of findings to community-based or outpatient populations. Inpatients generally have more severe illness, longer disease duration, and higher metabolic risk due to prolonged antipsychotic exposure and reduced physical activity. As a result, the observed SO prevalence may be higher, and the association between $SO_{ESPEN-M}$-defined SO and cognitive impairment in males may differ in less severe cohorts. Fourth, despite rigorous covariate adjustment, residual confounding from unmeasured variables (e.g., genetic susceptibility, neuroinflammation biomarkers like IL-6/TNF-α) remains possible. Future research should incorporate these mechanistic biomarkers to elucidate underlying pathways. Lastly, validating the $SO_{ESPEN-M}$ criteria across diverse ethnic populations and investigating its prognostic value for functional outcomes, such as disability and mortality, would improve its clinical applicability.

## Conclusion

In conclusion, this study identified a significant, independent association between SO, defined by the $SO_{ESPEN-M}$ criteria, and reduced cognitive performance in male inpatients with stable schizophrenia—an association not observed with

traditional SO$_{ESPEN}$ definitions. These findings support the clinical utility of BMI-adjusted diagnostic frameworks to better capture metabolically driven cognitive vulnerability in males and guide tailored interventions aimed at preserving both cognitive and neuromuscular function in schizophrenia care.

## Supporting information

**S1 Data. De-identified Participant-Level Data Including Demographic Variables, Clinical Measurements, MoCA-C Scores, and Covariates Analyzed in this Study.**
(XLSX)

## Acknowledgments

We wish to extend our deepest gratitude to all the patients who participated in this study, as well as to the staff for their unwavering support and commitment.

## Author contributions

**Conceptualization:** Yilin Wang, Youguo Tan.

**Formal analysis:** Yilin Wang, Youguo Tan.

**Funding acquisition:** Yan Guo, Jianfei Wu, Xiuping Lei, Hongli Zhang.

**Investigation:** Yan Guo, Jianfei Wu, Binyou Wang, Yu Liu, Maoya Xu, Yilin Wang.

**Methodology:** Xiuping Lei, Hongli Zhang, Yilin Wang, Youguo Tan.

**Project administration:** Yilin Wang.

**Resources:** Yan Guo, Jianfei Wu, Xiuping Lei.

**Writing – original draft:** Yan Guo, Jianfei Wu, Xiuping Lei, Hongli Zhang, Binyou Wang, Yu Liu, Maoya Xu, Yilin Wang.

**Writing – review & editing:** Yan Guo, Jianfei Wu, Binyou Wang, Yu Liu, Maoya Xu, Yilin Wang, Youguo Tan.

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
