## [Decision Letter · Decision Letter 0]

16 Jun 2025

PONE-D-25-22812The relationship between sarcopenic obesity and cognitive functionality among patients with stable schizophreniaPLOS ONE

Dear Dr. Wang,

Thank you for submitting your manuscript to PLOS ONE. After careful consideration, we feel that it has merit but does not fully meet PLOS ONE’s publication criteria as it currently stands. Therefore, we invite you to submit a revised version of the manuscript that addresses the points raised during the review process.

We look forward to receiving your revised manuscript.

Kind regards,

Ioannis Liampas, MD. PhD

Academic Editor

PLOS ONE

Journal Requirements:

“This study was supported by the Zigong Key Science and Technology Plan (Collaborative Innovation Project of Zigong Institute of Brain Sciences) (Grant Numbers: 2023-NKY-02-04, 2023-NKY-02-07, 2023-NKY-03-03), and the Collaborative Innovation Project of Zigong Medical Big Data and Artificial Intelligence Research Institute (Grant Number: 2024-YGY-02-04). The funder had no role in the design, data collection, data analysis, and reporting of this study.’

Reviewers' comments:

Reviewer's Responses to Questions

**Comments to the Author**

1. Is the manuscript technically sound, and do the data support the conclusions?

Reviewer #1: Yes

Reviewer #2: Yes

2. Has the statistical analysis been performed appropriately and rigorously? 

Reviewer #1: Yes

Reviewer #2: Yes

3. Have the authors made all data underlying the findings in their manuscript fully available?

Reviewer #1: No

Reviewer #2: Yes

4. Is the manuscript presented in an intelligible fashion and written in standard English?

Reviewer #1: Yes

Reviewer #2: Yes

5. Review Comments to the Author

Reviewer #1: Minor Comments

Writing clarity: Although the manuscript is well-written, a few sentences are overly long and could benefit from more concise phrasing (e.g., lines 247–255).

Terminology: Use consistent terminology that should be defined early and used uniformly.

Visuals: Consider adding a figure or flowchart to visualize the classification process or study design (e.g., participant selection flow).

Design: With clearly defined inclusion/exclusion criteria and the use of validated techniques (MoCA-C, SOESPEN, BIA), the process is robust. Although the cross-sectional aspect of the study is duly noted, the interpretation would be strengthened by a more thorough examination of the limitations of causation and possible reverse causality.

Reviewer #2: This cross-sectional study by Guo et al. examines the relationship between sarcopenic obesity (SO) and cognitive decline in stable inpatients with schizophrenia (N=228). SO was assessed using two diagnostic criteria assessments (SO- ESPEN and SO-ESPEN-M), which cognitive functioning was assessed using the Montreal Cognitive Assessment (MoCA-Chinese Version). Findings revealed that 17.1% of this sample met diagnostic criteria for SO, and that patients with SO had significantly lower MoCA scores than those without SO (and particularly so when defined using the SO-ESPEN-M criteria). This finding supports that early identification of SO phenotypes via body composition analysis and metabolic profiling is important for developing targeted interventions. The SO-ESPEN-M diagnostic criteria may be most relevant and applicable to patients with schizophrenia as it adjusts for BMI and provides a more accurate measure of muscle insufficiency, which, independent of BMI, is mechanistically associated with neuroinflammation and insulin resistance in patients with schizophrenia. Nonetheless, there are several confounding variables that limit the conclusions that can be drawn from this study. Below are some comments that would address some limitations/questions that remain:

1. Of the 90 patients (39.5%) that were female, how many were post-menopausal? Were subgroup analyses according to sex conducted? Given that SO is more prevalent in post-menopausal women, further stratifications according to age and sex would be essential.

2. Would it be possible to provide a breakdown of the most common antipsychotics used in this sample? It would be interesting to see if the risk of SO is highest with agents such as olanzapine and clozapine, similar to the risk of other metabolic disturbances. Furthermore, were any dose-related relationships observed?

3. Were other metabolic parameters available to include as covariates or potential predictors of cognitive scores (i.e., fasting glucose, insulin resistance)?

4. Did SO or cognition scores correlate with any measures of psychopathology? How about illness duration and cognition?

5. What was the duration of inpatient stay among these patients? How does the rate of SO in inpatients compare to that of age and sex-matched outpatients with schizophrenia (if this exists in the literature)? The generalizability of the study findings should be discussed in more detail.

6. If possible, it would be useful to see the data presented in tables 3 and 4 as graphs.

6. PLOS authors have the option to publish the peer review history of their article (what does this mean?). If published, this will include your full peer review and any attached files.

Reviewer #1: **Yes: **Mohammed Abdullah Aljaffer

Reviewer #2: No

---

## [Author Response · Author response to Decision Letter 1]

2 Jul 2025

General Reply

Many thanks to the Referee for encouraging our work and for giving useful comments

for clarifying, improving and correcting some materials in the paper.

Now we have carefully revised the paper according to your comments, as explained

below.

My colleagues and I are very grateful for your consideration of publishing our work in

PLoS One and look forward to hearing from you about the further disposition of this re-submission.

Specific reply to Technical comments

Reviewer: 1

Dear reviewer, Thank you for reviewing our manuscript and for the constructive comments, which greatly helped us to improve the manuscript. Considering another reviewer’s valuable advice, we have heavily revised the content of the article. The major revisions made to the study include: 1) Performing analyses stratified by sex; 2) Broadening our covariate set to include clinically relevant factors such as metabolic parameters, indices of insulin resistance, drug dosage, symptom scores, along with specific factors for female participants: age at menarche, menopausal status, and age; 3) Enhancing the clarity of critical data through improved visualizations, facilitating easier interpretation and comparison; 4) Re-analyzing the data based on the updated study design, which required thorough revisions to the Methods, Results, and Discussion sections, including a deeper discussion informed by the updated results. We were pleased to receive the encouraging comments about our study. The manuscript has been carefully revised, and our point-by-point responses to your comments are detailed below. We hope we have successfully addressed all your concerns.

Point 1

Referee: Writing clarity: Although the manuscript is well-written, a few sentences are overly long and could benefit from more concise phrasing (e.g., lines 247–255).

Reply: Thank you for your valuable feedback regarding the writing clarity of our manuscript. In response to this comment, we have carefully reviewed the manuscript once again, paying close attention to sentence structure and conciseness throughout the text. The specific sentences mentioned (lines 247-255) and other potentially wordy passages have been revised for greater clarity. To ensure the language meets the highest standards of clarity and readability for an English-speaking audience, we have submitted the revised manuscript to a native English speaker with expertise in scientific writing for a final proofread and language polishing. Their feedback has been incorporated into the version you will receive. We believe these revisions, now presented in the attached revised manuscript, have significantly improved the overall flow and clarity, directly addressing the specific points you raised. We hope this revised version now meets your expectations.

Point 2

Referee: Terminology: Use consistent terminology that should be defined early and used uniformly.

Reply: Thank you for pointing out the importance of consistent terminology. We recognize that clear and uniform language is vital for ensuring our findings are easily understood by the readership. To address this, we have meticulously revised the manuscript. We have defined key terms early in the text, specifically in the Introduction and Methods section, and have made certain that these terms are used consistently throughout the Results and Discussion. This approach aims to provide a clear framework for interpreting our study. We believe the revised manuscript now presents the information more clearly and consistently, as per your suggestion.

Point 3

Referee: Visuals: Consider adding a figure or flowchart to visualize the classification process or study design (e.g., participant selection flow).

Reply: Thank you very much for your constructive suggestion regarding the visuals in our manuscript. We appreciate you highlighting the potential benefit of adding figures to enhance clarity, specifically for visualizing the classification process or study design, including participant selection flow. Following your guidance, we have incorporated your suggestion. In the revised manuscript, we have added a flowchart detailing the patient selection process (Figure 1), which we hope provides a clearer overview of how participants were included and excluded. Furthermore, to better illustrate the differences between groups, we have also included Figure 2, which compares MoCA-C scores between the SO and non-SO groups defined by different criteria in both males and females. Additionally, we have used a forest plot to present the results of the multiple linear regression analysis (Figure 3), improving the presentation of the data. We believe these additions significantly enhance the readability and visual appeal of the manuscript. We hope that your comment has been effectively addressed.

Point 4

Referee: Design: With clearly defined inclusion/exclusion criteria and the use of validated techniques (MoCA-C, SOESPEN, BIA), the process is robust. Although the cross-sectional aspect of the study is duly noted, the interpretation would be strengthened by a more thorough examination of the limitations of causation and possible reverse causality.

Reply: Thank you for your insightful comment regarding the limitations of our cross-sectional study design, particularly concerning causal inference and potential reverse causality. We agree that the cross-sectional nature of our study inherently limits our ability to draw conclusions about causation. To address this point, we have significantly expanded the discussion of limitations in the revised manuscript. Specifically, we have added the following paragraph to the ‘Limitations’ section: “This study has several limitations that warrant consideration. First, its cross-sectional design, although based on validated methods (e.g., MoCA-C, SOESPEN criteria, BIA), precludes causal inference between SO and cognitive impairment. Longitudinal studies are necessary to determine temporal relationships and evaluate whether SO precedes or results from cognitive decline”. This addition clearly acknowledges the limitation regarding causality and outlines the need for future longitudinal research to determine the temporal relationship between SO and cognitive decline. We hope this modification adequately addresses your concern. Thank you again for your valuable feedback.

Reviewer: 2

Point 1

Referee: Of the 90 patients (39.5%) that were female, how many were post-menopausal? Were subgroup analyses according to sex conducted? Given that SO is more prevalent in post-menopausal women, further stratifications according to age and sex would be essential.

Reply: Thank you very much for your insightful comments regarding the demographic characteristics of our study population, particularly concerning the female participants and the potential influence of menopausal status on SO. We appreciate you highlighting these important points, which are indeed constructive and warrant careful consideration. As you correctly pointed out, factors such as sex and age can significantly influence the occurrence of SO. Following your guidance, we have made the following revisions to our manuscript:

1, Subgroup analysis by sex: We have conducted stratified analyses according to sex, as suggested. This allows us to examine the relationships within male and female participants separately.

2, Inclusion of menopausal status: Recognizing the higher prevalence of SO in post-menopausal women, we have incorporated menopausal status (including age at menopause for post-menopausal women) and age at menarche as covariates in our statistical models. This helps control for potential confounding effects related to hormonal changes associated with the menopause transition.

Regarding your suggestion for further stratifications according to age, we acknowledge the value of this approach. However, due to limitations in the existing literature providing clear guidance on specific age cut-offs for stratification in this context, coupled with constraints on our sample size, performing extensive age-based subgroup analyses was not feasible in this revision.

To reflect these changes, we have made substantial adjustments to the presentation of our results. Please find the updated tables and figures in the revised manuscript:

Revised Table 1: Now includes detailed participant characteristics, reflecting the updated covariates.

Revised Table 2: Presents comparisons between SO and non-SO groups according to ESPEN/EASO criteria and modifications, now including results for the overall population as well as stratified by sex.

Revised Table 3: Shows univariate analyses of general information and MoCA-C scores in the patient.

New Figure 1: Illustrates the participant selection flowchart.

New Figure 2: Visualizes the comparison of MoCA-C scores between SO and non-SO groups defined by different criteria in males and females.

New Figure 3: Presents a forest plot displaying the results of the multivariate linear regression analysis for factors associated with MoCA-C scores.

Moreover, We have also revised the limitations section in the Discussion. Specifically, we have expanded the second point to acknowledge the challenges posed by our single-center design, sample size, and medication heterogeneity. We now explicitly state that the modest sample size limited our ability to perform the requested age stratification and also precluded a detailed analysis of specific antipsychotic drug effects. The updated limitations section reads as follows (relevant part): “Second, the single-center design and limited sample size imposed constraints. Specifically: 1) The sample size, compounded by the scarcity of research on this specific issue in schizophrenia patients, precluded further stratified analyses by specific age groups. Although age and menopausal status were included as covariates, statistical power was insufficient to detect potentially nuanced age-related effects or interactions within narrower age strata. 2) Substantial heterogeneity in antipsychotic medication regimens among participants – with over ten distinct monotherapies or bivalent combinations represented – prevented robust evaluation of potential differential effects of specific antipsychotic types on SO or cognitive outcomes”.

We trust that these revisions adequately address your concerns and provide a more nuanced understanding of the relationships between SO, cognitive function, and sex-related factors within our study population.

Thank you again for your valuable feedback, which has significantly strengthened our manuscript.

Point 2

Referee: Would it be possible to provide a breakdown of the most common antipsychotics used in this sample? It would be interesting to see if the risk of SO is highest with agents such as olanzapine and clozapine, similar to the risk of other metabolic disturbances. Furthermore, were any dose-related relationships observed?

Reply: Thank you very much for your further insightful questions regarding the antipsychotic medication use in our sample and the potential dose-related relationships. Regarding your request for a breakdown of the most common antipsychotics used and the potential risk associated with specific agents like olanzapine and clozapine, we acknowledge the significance of this inquiry. As we mentioned in our response to your previous comments and have now explicitly stated in the revised manuscript’s discussion section under limitations, the heterogeneity in antipsychotic medication use within our sample was considerable. We observed over ten distinct monotherapy or bivalent combination regimens. Given the relatively modest overall sample size of our single-center study, we lacked the necessary statistical power to perform meaningful subgroup analyses based on specific antipsychotic types. Therefore, we were unable to provide a detailed breakdown or assess differential risks associated with individual agents, such as comparing olanzapine or clozapine directly to other antipsychotics regarding SO risk, similar to what is known for other metabolic disturbances. We fully agree that this is a highly relevant and worthwhile area for investigation. We have added the relevant limitation to the discussion, stating: “...2) Substantial heterogeneity in antipsychotic medication regimens among participants – with over ten distinct monotherapies or bivalent combinations represented – prevented robust evaluation of potential differential effects of specific antipsychotic types on SO or cognitive outcomes.” We concur that this limitation significantly restricts our ability to address your specific question about agent-specific risks in this current study.

However, we recognize the importance of considering medication dosage. In response to this, and to partially address the influence of medication load, we have incorporated the “Chlorpromazine equivalent dose” as a covariate in our analyses during the revision process. The results incorporating this covariate are presented in the revised manuscript, and we invite you to review the relevant sections for these findings.

We view this as an important direction for future research. We plan to collaborate with multiple centers to enroll a larger and more diverse patient population in subsequent studies. This will provide the statistical power needed to conduct the stratified analyses you suggest, allowing for a more detailed examination of the risks associated with specific antipsychotic agents and potentially dose-related effects on SO in patients with schizophrenia. We are particularly grateful to you for highlighting this important and promising research avenue.

Thank you again for your valuable feedback, which has significantly strengthened our manuscript.

Point 3

Referee: Were other metabolic parameters available to include as covariates or potential predictors of cognitive scores (i.e., fasting glucose, insulin resistance)?

Reply: Thank you very much for your insightful suggestion regarding the inclusion of additional metabolic parameters as covariates or potential predictors of cognitive scores. We greatly appreciate your professional guidance, which has significantly enhanced the depth of our analysis. Following your recommendation, we have incorporated relevant metabolic and insulin resistance indicators into our revised analyses. Specifically, we included fasting blood glucose, Alanine Aminotransferase (ALT)/Aspartate Aminotransferase (AST), Uric acid, and the ratio of Triglyceride to High-Density Lipoprotein Cholesterol (Triglyceride/HDL-C). To reflect this update, we have modified the description of the covariates in the Methods section. The revised text now states: “A comprehensive set of covariates was systematically collected from multiple data sources. Demographic information, including age, sex, education level (illiterate/high school and below/university and above), marital status (married/unmarried/divorced/widowed), number of siblings, number of children, and smoking and drinking history, was obtained from clinical records. Other clinical variables included vision and hearing impairments, disease duration, hospitalized time, family psychiatric history, first-episode status, fall history, COVID-19 history, number of chronic comorbidities, current antipsychotic regimen, chlorpromazine equivalent dose, and relevant metabolic and insulin resistance markers (fasting blood glucose; alanine aminotransferase [ALT]/aspartate aminotransferase [AST] ratio; uric acid; triglyceride[TG]/high-density lipoprotein cholesterol [HDL-C] ratio). Psychological variables included depression symptoms assessed using the Patient Health Questionnaire-9 (PHQ-9) [30], anxiety symptoms using the Generalized Anxiety Disorder-7 scale (GAD-7) [31], and psychotic symptoms using the Positive and Negative Syndrome Scale (PANSS) [32].”.

Furthermore, the interpretation of the data and the corresponding analysis sections in the Results have been updated accordingly. Please refer to Revised Table 1 and Revised Table 3 for the updated participant characteristics and univariate analysis results, and to the revised Figure 3 for the updated forest plot of the multivariate linear regression analysis. Detailed information can be found in the revised manuscript.

Thank you again for your valuable feedback

---

## [Decision Letter · Decision Letter 1]

1 Aug 2025

The relationship between sarcopenic obesity and cognitive functionality among inpatients with stable schizophrenia

PONE-D-25-22812R1

Dear Dr. Wang,

We’re pleased to inform you that your manuscript has been judged scientifically suitable for publication and will be formally accepted for publication once it meets all outstanding technical requirements.

Kind regards,

Ioannis Liampas, MD. PhD

Academic Editor

PLOS ONE

Reviewers' comments:

Reviewer's Responses to Questions

**Comments to the Author**

1. If the authors have adequately addressed your comments raised in a previous round of review and you feel that this manuscript is now acceptable for publication, you may indicate that here to bypass the “Comments to the Author” section, enter your conflict of interest statement in the “Confidential to Editor” section, and submit your "Accept" recommendation.

Reviewer #2: All comments have been addressed

2. Is the manuscript technically sound, and do the data support the conclusions?

Reviewer #2: Yes

3. Has the statistical analysis been performed appropriately and rigorously? 

Reviewer #2: Yes

4. Have the authors made all data underlying the findings in their manuscript fully available?

Reviewer #2: Yes

5. Is the manuscript presented in an intelligible fashion and written in standard English?

Reviewer #2: No

6. Review Comments to the Author

Reviewer #2: I am satisfied with the revisions and have no further comments. I would recommend the manuscript for publication.

7. PLOS authors have the option to publish the peer review history of their article (what does this mean?). If published, this will include your full peer review and any attached files.

Reviewer #2: No

---

## [Editor Report · Acceptance letter]

PONE-D-25-22812R1

PLOS ONE

Dear Dr. Wang,

I'm pleased to inform you that your manuscript has been deemed suitable for publication in PLOS ONE. Congratulations! Your manuscript is now being handed over to our production team.

Kind regards,

on behalf of

Dr. Ioannis Liampas

Academic Editor

PLOS ONE